# Effect of COVID-19 Pandemic on the Change in Skeletal Muscle Mass in Older Patients with Type 2 Diabetes: A Retrospective Cohort Study

**DOI:** 10.3390/ijerph18084188

**Published:** 2021-04-15

**Authors:** Yuka Hasegawa, Fuyuko Takahashi, Yoshitaka Hashimoto, Chihiro Munekawa, Yukako Hosomi, Takuro Okamura, Hiroshi Okada, Takafumi Senmaru, Naoko Nakanishi, Saori Majima, Emi Ushigome, Masahide Hamaguchi, Masahiro Yamazaki, Michiaki Fukui

**Affiliations:** Department of Endocrinology and Metabolism, Graduate School of Medical Science, Kyoto Prefectural University of Medicine, Kyoto 602-8566, Japan; yuka-h@koto.kpu-m.ac.jp (Y.H.); fuyuko-t@koto.kpu-m.ac.jp (F.T.); c-mori@koto.kpu-m.ac.jp (C.M.); hy0226@koto.kpu-m.ac.jp (Y.H.); d04sm012@koto.kpu-m.ac.jp (T.O.); conti@koto.kpu-m.ac.jp (H.O.); semmarut@koto.kpu-m.ac.jp (T.S.); naoko-n@koto.kpu-m.ac.jp (N.N.); saori-m@koto.kpu-m.ac.jp (S.M.); emis@koto.kpu-m.ac.jp (E.U.); mhama@koto.kpu-m.ac.jp (M.H.); masahiro@koto.kpu-m.ac.jp (M.Y.); michiaki@koto.kpu-m.ac.jp (M.F.)

**Keywords:** muscle mass, sarcopenia, lifestyle, pandemic, older, type 2 diabetes

## Abstract

*Background*: The aim of this study was to investigate the effect of the coronavirus disease (COVID-19) pandemic restrictions on the change in muscle mass in older patients with type 2 diabetes (T2D), who were not infected with COVID-19. *Methods:* In this retrospective cohort study, data were obtained from outpatients who underwent bioelectrical impedance analysis at least twice before April 2020 and at least once thereafter. Skeletal muscle mass index (SMI, kg/m^2^) was calculated as appendicular muscle mass (kg) divided by height squared (m^2^). Change in SMI (kg/m^2^/year) was calculated as (follow-up SMI—baseline SMI/follow-up period). The differences between the changes in SMI before and after the start of the COVID-19 pandemic were evaluated using paired *t* test. **Results:** This study recruited 56 patients, with a mean (SD) age of 75.2 (7.1) years. SMI changed from 6.7 (0.9) to 6.8 (0.9) kg/m^2^ before the COVID-19 pandemic, whereas SMI changed from 6.8 (0.9) to 6.6 (0.9) kg/m^2^ after the start of the COVID-19 pandemic. SMI decreased after the start of the COVID-19 pandemic compared with before the pandemic (−0.117 (0.240) vs. 0.005 (0.289) kg/m^2^/year, *p* = 0.049). This decrease was observed in men (−0.159 (0.257) vs. 0.031 (0.325) kg/m^2^/year, *p* = 0.038), patients with poor glycemic control (−0.170 (0.264) vs. 0.031 (0.285) kg/m^2^/year, *p* = 0.042), and those with a long diabetes duration (−0.153 (0.229) vs. 0.082 (0.291) kg/m^2^, *p* = 0.049). *Conclusions:* The COVID-19 pandemic restrictions caused muscle mass loss in older patents with T2D. Actions, including recommendation of exercise and adequate diet intake, are needed to prevent loss of muscle mass.

## 1. Introduction

The coronavirus disease (COVID-19) pandemic has had a severe impact on public health worldwide [1]. In early January 2021, more than 80 million confirmed cases and 1.8 million deaths due to COVID-19 had been reported in 222 countries [2]. In response to the pandemic, several countries, such as Italy and France, implemented city lockdowns [3]. In Japan, a state of emergency, with request-based measures of encouraging the populace to remain at home, was declared in seven major cities, including Tokyo and Osaka, on 7 April 2020, and this progressed to a nationwide lockdown on 16 April 2020 [4]. Thus, along with its spread, the COVID-19 pandemic has led to a dramatic change in lifestyles, such as unhealthy dietary patterns, increased alcohol consumption, increased smoking frequency, and reduced physical activity [5,6,7]. During the COVID-19 pandemic, our research group conducted a study of patients with type 2 diabetes (T2D) and reported lifestyle changes as a result of the COVID-19 pandemic, including increased stress, increased total diet intake, increased snack consumption, decreased sleep duration, and reduced physical activity [8].

Muscle is a major organ for glucose metabolism [9]; therefore, muscle is an important target for treatment of patients with T2D. Patients with diabetes have accelerated muscle catabolism because of insulin signal attenuation and insulin resistance [10]. Moreover, older patients with T2D often have sarcopenia [11]. Therefore, for older patients with T2D, prevention of muscle mass reduction is an important goal.

Several questionnaire studies in Asian, African, and European countries reported a decrease in physical activity levels and an increase in sitting time due to measures put in place to reduce the spread of COVID-19 [12,13]. A decrease in moderate-to-vigorous physical activity in people with chronic diseases during the lockdown period has also been reported [14]. These results show that measures put in place to reduce the COVID-19 pandemic reduce incidental and planned physical activities. Taking these findings together, COVID-19 pandemic restrictions during the pandemic lockdown or periods of staying at home caused increased stress, changes in dietary patterns, and decreased physical activity. To the best of our knowledge, there are no reports on the effect of the COVID-19 pandemic restrictions on change in muscle mass, although several reviews have shown the possibility of muscle mass loss with the spread of the disease, especially among older people [4,15,16,17,18]. Thus, the primary aim of this retrospective cohort study was to investigate the effect of COVID-19 pandemic restrictions on the change in muscle mass in older patients with T2D who were not infected with COVID-19, and the secondary aim was to clarify the risk factors for the change in muscle mass during COVID-19 pandemic restrictions.

## 2. Materials and Methods

### 2.1. Participants and Study Design

The KAMOGAWA cohort study, which included the outpatient clinic at Kyoto Prefectural University of Medicine (KPUM) (Kyoto, Japan) and the Kameoka Municipal Hospital (Kameoka, Japan), has been conducted since 2014 among patients with diabetes [19]. This study was a retrospective cohort study. Medical data were collected as anonymized data after obtaining informed consent, and data of bioelectrical impedance analysis (BIA), lifestyle, medications, and laboratory data were obtained from the participants who underwent BIA at least twice before April 2020 and at least once thereafter in this study. In Japan, the first state of emergency was declared on 7 April 2020, covering seven prefectures, and on 16 April, the state of emergency was extended nationwide. Thus, the date of the COVID-19 outbreak in this study was set as 16 April. Point 2 was the last BIA before the COVID-19 pandemic, point 1 was the BIA performed before point 2, and point 3 was the BIA performed after the start of the pandemic (Figure 1). The exclusion criteria were as follows: (1) incomplete data, (2) data of point 3, the last BIA, collected before June 2020, (3) data collection period between point 1 and point 2 and/or point 2 and point 3 of less than 6 months, (4) data collection period between point 2 and point 3 of more than 2.5 years, (5) non-elderly (age under 60 years old) patients [20], (6) patients with renal failure [21], and (7) patients with cardiac failure, defined by brain natriuretic peptide ≥100 pg/mL [22]. This cohort study was permitted by the Ethics Committee of KPUM (No. RBMR-E-466-5).

### 2.2. Measurement of Body Composition

Body composition of the participants was evaluated using a multifrequency impedance BIA, InBody 720 (InBody Japan, Tokyo, Japan) [23], which has good correlation with dual-energy X-ray absorptiometry. Using this analyzer, data on body weight (BW, kg), appendicular muscle mass (kg), and fat mass (kg) were collected. Body mass index (BMI, kg/m^2^) was calculated as BW (kg) divided by height squared (m^2^), while skeletal muscle mass index (SMI, kg/m^2^) was determined by appendicular muscle mass (kg) divided by height squared (m^2^) [24]. Percent fat mass (%) was calculated as fat mass (kg)/BW (kg) ×100. Change in SMI (kg/m^2^/month) was calculated as (follow-up SMI [kg/m^2^]—baseline SMI [kg/m^2^])/follow-up period (year). Similarly, changes in BW, appendicular muscle mass, and body fat and percent body fat were also defined. Obesity was defined as a BMI of ≥25 kg/m^2^ [25]. In addition, low muscle mass was defined as SMI <7.0 kg/m^2^ in men and <5.7 kg/m^2^ in women [24].

### 2.3. Lifestyle, Medications, and Laboratory Data Collection Participants and Study Design

According to a self-administered questionnaire, participants were classified into non-smokers or current smokers and non-habitual or habitual exercisers if they engaged in any of the physical activities at least once a week [26] and non-alcohol or alcohol consumers [26]. Data on medications, including medications for diabetes, hypertension, and hyperlipidemia, and blood samples, including creatinine and brain natriuretic peptide, which were collected in the morning for biochemical measurements, were obtained from the medical records at point 2. In addition, the data of hemoglobin A1c (HbA1c) were collected at all points. The estimated glomerular filtration rate (eGFR; mL/min/1.73 m^2^) was calculated using the Japanese Society of Nephrology equation (eGFR = 194 × serum creatinine^−1.094^ × age^−0.287^ × (0.739 for women)) [27]. Renal failure was defined as an eGFR of <30 mL/min/1.73 m^2^ [21].

### 2.4. Questionnaire to Assess Change in Stress Levels and Lifestyle Due to the COVID-19 Pandemic

To assess stress or lifestyle changes due to the COVID-19 pandemic, we asked some patients a number of simple questions from 16 April to 31 May 2020. The details of this questionnaire were presented in a previous study [8]. The questionnaire consists of short questions regarding stress, sleep duration, exercise levels, and changes in lifestyle due to the pandemic. A visual analog scale (VAS; 0 = considerably reduced, 5 = no change, and 10 = considerably increased) was used, and we classified patients into the following categories: increased stress (VAS ≥ 6), decreased sleep duration (VAS ≤ 4), and decreased exercise (VAS ≤ 4).

### 2.5. Calculation of Sample Size

Since there are no data of the sample size from previous studies, we set the detection power at 0.8 and the standard deviation of difference at 0.5. Using R (ver. 4.0.3; The R Foundation for Statistical Computing, Vienna, Austria), we calculated the sample size, which indicted a minimum of 33.672 participants.

### 2.6. Statistical Analyses

JMP version 13.2 software (SAS Institute, Cary, NC, USA) was used for the statistical analyses. Results are presented as means (standard deviation, SD) and frequencies (percentage). A *p* value < 0.05 was considered statistically significant.

The differences among three points of measurement were evaluated by repeated measures analysis of variance (ANOVA) with Bonferroni correction. The differences in the changes observed in SMI and the other variables before and after the start of the COVID-19 pandemic were evaluated using the paired t test. Subsequently, these differences were investigated according to their status (yes/no) in the following subgroups: sex, exercise habit, smoking, habitual alcohol consumption, HbA1c > 7.0%, obesity, and duration of diabetes of ≥20 years.

Repeated measures ANOVA was performed to investigate the effect of the COVID-19 pandemic on change in SMI adjusting for age, sex, exercise habit, smoking, habitual alcohol consumption, HbA1c, and duration of diabetes.

Furthermore, the data of patients who answered the questions about changes in stress levels and lifestyle due to the COVID-19 pandemic were also summarized, and we investigated the differences in the changes in SMI before and after the start of the pandemic using the paired t test according to patient stress levels and exercise status.

## 3. Results

Among 538 patients with T2D who participated in the KAMOGAWA-DM cohort study, we extracted data on 105 patients who had undergone BIA at least three times. Of the 105, we excluded 48 patients who met the exclusion criteria (Figure 2): incomplete data (1 patient), data collected at point 3 before June 2020 (27 patients), data collection period between point 1 and point 2 and/or point 2 and point 3 of less than 6 months (4 patients), data collection period between point 2 and point 3 of more than 2.5 years (7 patients), age under 60 years old (6 patients), renal failure (one patient). None of the patients had cardiac failure. Thus, 56 patients were finally included in this study.

Baseline characteristics, point 2, of the study participants are shown in Table 1. Mean (SD) age, body weight, and duration of diabetes were 75.2 (7.1) years, 59.3 (10.3) kg and 19.7 (8.2) years, respectively. The proportions of men, smokers, exercisers, and alcohol consumers were 62.5%, 44.6%, 60.7%, and 46.4%, respectively. Among the participants, 22 patients answered the questions regarding the change in stress levels or lifestyle due to the COVID-19 pandemic (Table 1).

Changes in body composition before and after the start of the COVID-19 pandemic are shown in Table 2. Both SMM and SMI significantly decreased after the start of the pandemic compared with before the pandemic (SMM, *p* = 0.047 and SMI, *p* = 0.049). In addition, the proportion of low muscle mass was not changed from point 1 (*n* = 18) to point 2 (*n* = 18 but increased from point 2 (*n* = 18) to point 3 (*n* = 22). HbA1c was not changed due to COVID-19 pandemic restrictions. On the other hand, the proportions of usage of insulin sensitizers and SGLT2 inhibitors were increased with COVID-19 pandemic restrictions.

The results of the subgroups analyses on the change in SMI before and after the start of the COVID-19 pandemic are shown in Table 3. Interestingly, SMI significantly decreased after the start of the pandemic compared with before the pandemic in men, patients with poor glycemic control, and those with a long diabetes duration.

Table 4 shows the effect of the COVID-19 pandemic on change in SMI. The COVID-19 pandemic was independently associated with change in SMI (η^2^ = 0.108, *p* = 0.029).

Although there were few participants who answered the questions regarding the change in stress levels or lifestyle due to the COVID-19 pandemic, SMI significantly decreased after the start of the pandemic compared with the SMI before the pandemic in patients with decreased sleep duration (*n* = 4) and decreased exercise participation (*n* = 11) (Table 5).

## 4. Discussion

In this study we observed a significant decrease in SMI in older patients with T2D, who were not infected with COVID-19, due to the COVID-19 pandemic restrictions, compared with the SMI before the pandemic, suggesting that the pandemic restrictions caused muscle mass loss.

Muscle is a major organ for glucose metabolism [9]; therefore, muscle is an important target for treatment of patients with T2D. It has been reported that muscle mass loss is associated with low quality of life [28], cardiovascular disease [29,30], and mortality [31]. Previous studies revealed that muscle mass of patients with T2D is unchanged or slightly decreased in the natural course [32,33,34]. In this study, muscle mass was unchanged before the pandemic. On the other hand, muscle mass was decreased after the start of the pandemic. This might be due to the COVID-19 pandemic restrictions. It has been reported that poor glycemic control results in more muscle mass loss [32,33,34] and that among older patients with T2D, the decrease in muscle mass compared to non-diabetic people is more remarkable in men [35]. Moreover, the longer the duration of diabetes, the more diabetic complications develop, accelerating muscle weakness and loss of muscle mass [35,36].

Actions such as social distancing, isolation, and house arrest with self-quarantine during the COVID-19 pandemic resulted in several restrictions on activities of daily living, although these actions might have been essential to control the spread of COVID-19. There have been several reports on decreased physical activities during the pandemic. A questionnaire study on 1047 participants from Asia, Africa, and Europe, which included several healthy individuals, reported that during the COVID-19 pandemic, exercise duration and physical activities decreased while sitting time increased [12]. Similarly, a study on chronically ill patients in Spain found a significant decline in moderate-to-vigorous physical activities, regardless of the sex, in the 18–44 and 55–64 age groups [14]. A questionnaire survey of 72 patients with T2D in Spain also revealed a significant increase in sedentary time and a decrease in walking and moderate physical activity time among study participants during the pandemic compared to before the pandemic [37]. Measurements of the time spent at home and distance traveled before and after the start of lockdown using smartphone data from 1062 participants in five European countries showed that younger people spent more time at home during the lockdown than older people, and younger people took fewer steps per day [38]. Although there was no significant difference in the time spent at home between the high and low BMI groups, those in the low BMI group spent more time walking than those in the high BMI group [38]. Physically inactive older adults have been found to have an increased risk of all-cause and cardiovascular mortalities, fractures, decreased activities of daily living, functional limitations, risk of falls, cognitive decline, dementia, and depression compared with active older adults [39]. Furthermore, a meta-analysis showed that sedentary time is associated with higher risk of all-cause mortalities and those from cardiovascular diseases [40]. On the contrary, higher physical activity levels tended to reduce the risk of death from longer sedentary periods [41].

In particular, the reduction in physical activity in diabetic patients may have more serious consequences for diabetic patients during COVID-19 pandemic [42]. Previous studies have shown that maintaining negative lifestyle habits such as decreased physical activity, sedentary behavior, and consumption of unhealthy foods for several weeks can cause impaired glycemic control (increased insulin resistance) and increased total body fat and abdominal fat [43]. Adequate glycemic control is associated with a significant reduction in mortality and disease complications in COVID-19 diabetic patients [44].

Thus, the need for strength training has been widely proposed, especially in at-risk populations such as the elderly, those with T2D, and those with respiratory and cardiovascular diseases who need to stay healthy during the lockdown period [45,46,47]. Countries have also proposed the need to maintain adequate physical activities and specific exercise programs during this period [12,45,46,48]. To increase the awareness on frailty prevention during a pandemic, the Japanese Geriatrics Society published a report suggesting the reduction of sitting time, exercising to maintain muscle strength, and walking [48].

The risk of death from COVID-19 is higher among the elderly [49], men [50], those with poor management of diabetes [42], and those with long diabetes duration [51], while smoking is associated with the progression of COVID-19 [52] even though these patients may have been more conscious of the social distancing guidelines and reduced contact with others. Moreover, older age [53], diabetes, and smoking [54] were reported as risk factors for sarcopenia, and the decrease in physical activities may have caused excessive muscle mass loss in those at a high risk of sarcopenia. Therefore, it is necessary to recommend exercise and adequate protein intake to those at a higher risk of sarcopenia to prevent muscle mass loss [55].

Preventive care training for older adults is a common practice in Japan. Several local governments in Japan invested in improving the environment of older adults in their communities, which led to the organization of community exercises for the elderly to prevent functional decline and falls and enabled them to be functionally active before the COVID-19 pandemic [3]. These became dormant during the pandemic, which may have contributed to the severe muscle weakness observed among the elderly, especially in men. Contrarily, many older women in Japan were responsible for household chores and may not have been actively engaged in outdoor activities on a regular basis, which may have contributed to the low muscle mass loss.

In addition, HbA1c was not changed due to COVID-19 pandemic restrictions. On the other hand, the proportions of usage of insulin sensitizers and SGLT2 inhibitors were increased with COVID-19 pandemic restrictions. These results suggest that glycemic control could have been achieved by adjusting medications.

Although the sample size used in this study is small, we also found a decrease in SMI in patients with increased stress levels, decreased sleep duration, and decreased exercise participation due to the COVID-19 pandemic in a questionnaire survey. Stress level and sleep duration have been reported to be associated with sarcopenia in previous studies. There was a negative association between grip strength and stress levels in older people [56]. There was also an association between sleep disturbance and sarcopenia in older patients with diabetes [57].

There are several limitations in this study. First, because this was an observational study, we were unable to examine the effects of other factors, including energy intake, on muscle mass during the study. Second, we were unable to quantitatively assess changes in engagement in physical activity, which could reasonably explain the decreased physical activity levels during periods of staying at home. Third, there is a possibility that the difference in the period between from point 1 to point 2 and point 2 to point 3 (*p* < 0.001, by paired *t* test) might affect the decline of SMI before and after the state of emergency, although we corrected the effect of the difference in the period by determining the rate of change in SMI per year. Fourth, functional measures, such as hand grip strength, sit to stand test, and timed walking test are important markers. Unfortunately, however, we did not have the data of these functional measures. Fifth, the number of included patients is low compared to the entire study population. This might act as a confounder. Lastly, the participants in this study were predominantly older Japanese patients, and it is unclear whether the study can be extrapolated to a younger population.

## 5. Conclusions

This study revealed that the COVID-19 pandemic restrictions caused muscle mass loss in older patents with T2D. A decrease in SMI after the start of the pandemic was especially remarkable among men, patients with poor glycemic control, and those with a long diabetes duration. Actions, including recommendation of exercise, are needed to prevent loss of muscle mass. Further studies including exercise intervention would be valuable.

## Figures and Tables

**Figure 1 ijerph-18-04188-f001:**
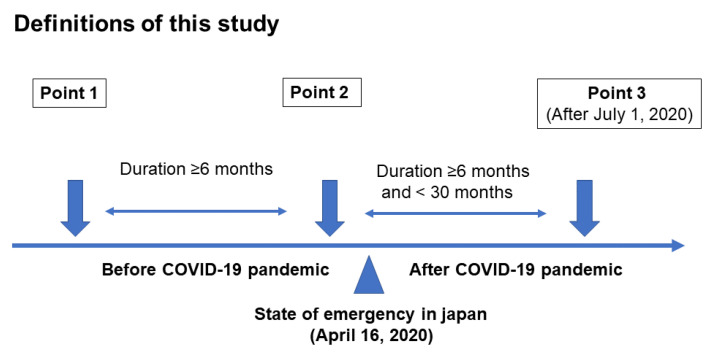
Definitions of this study.

**Figure 2 ijerph-18-04188-f002:**
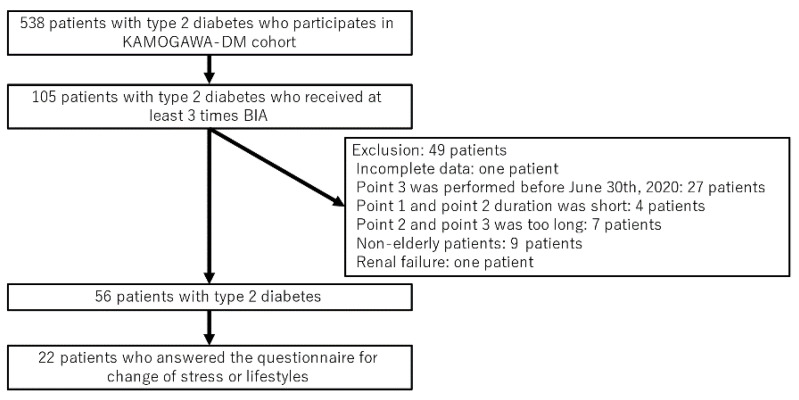
Inclusion and exclusion flow.

**Table 1 ijerph-18-04188-t001:** Baseline characteristics (point 2) of study participants.

N = 56	Point 1	Point 2	Point 3
Age, years	-	75.2 (7.1)	-
Men	-	35 (62.5)	-
Duration of diabetes, years	-	19.7 (8.2)	-
Smokers	-	25 (44.6)	-
Exercisers	-	34 (60.7)	-
Alcohol consumers	-	26 (46.4)	-
Height, cm	-	160.5 (8.2)	-
HbA1c, %	7.2 (0.8)	7.1 (0.7)	7.1 (0.7)
HbA1c, mmol/mol	55.1 (9.1)	54.4 (7.9)	54.4 (7.9)
Insulin secretagogues	49 (86.0)	49 (86.0)	51 (89.5)
Insulin sensitizers	23 (40.4)	24 (42.1)	28 (49.1)
α-glucosidase inhibitors	11 (19.3)	11 (19.3)	13 (22.8)
Sodium glucose cotransporter 2 inhibitors	10 (17.5)	16 (28.1)	19 (33.3)
GLP-1 receptor agonists	3 (5.2)	4 (7.0)	5 (8.8)
Insulin	14 (24.6)	14 (24.6)	15 (26.3)
Renin-angiotensin system inhibitors	-	34 (59.6)	-
Calcium channel blockers	-	19 (33.3)	-
Other antihypertension drug	-	15 (26.3)	-
Medication for dyslipidemia	-	33 (57.8)	-
**N = 22**			
Age, years	-	75.3 (6.3)	-
Men	-	15 (68.1)	-
Duration of diabetes, years	-	21.2 (9.8)	-
Smokers	-	14 (63.6)	-
Exercisers	-	14 (63.6)	-
Alcohol consumers	-	10 (45.4)	-
Height, cm	-	155.1 (24.7)	-
HbA1c, %	-	7.4 (0.7)	-
HbA1c, mmol/mol	-	57.4 (7.6)	-
Insulin secretagogues	-	22 (100.0)	-
Insulin sensitizers	-	14 (63.6)	-
α-Glucosidase inhibitors	-	9 (40.9)	-
Sodium glucose cotransporter 2 inhibitors	-	7 (31.8)	-
GLP-1 receptor agonists	-	1 (4.5)	-
Insulin	-	6 (27.2)	-
Renin-angiotensin system inhibitors	-	15 (68.1)	-
Calcium channel blockers	-	3 (13.6)	-
Other antihypertension drug	-	5 (22.7)	-
Medication for dyslipidemia	-	13 (59.0)	-
Stress increased	-	5 (22.7)	-
Sleep duration decreased	-	4 (18.1)	-
Exercise decreased	-	11 (50.0)	-

Insulin secretagogues include sulfonylurea, dpp4 inhibitors, and glinide. Insulin sensitizers include biguanides and thiazolidine. Other antihypertension drugs include α-blockers, β-blockers, and diuretics. Medication for dyslipidemia included statin, fibrate, eicosapentaenoic acid, and ezetimibe. Data are expressed as means (standard deviation) and frequencies (percentage). GLP-1, glucagon like peptide-1. Among the participants, 22 patients answered the questions regarding the change in stress levels or lifestyle due to the COVID-19 pandemic.

**Table 2 ijerph-18-04188-t002:** Changes in body composition before and after the start of the COVID-19 pandemic.

	Point 1	Point 2	Point 3	* *p* Value	Change between Point 1 and Point 2	Change between Point 2 and Point 3	** *p* Value
Body weight, kg	59.3 (10.3)	59.1 (10.2)	58.1 (10.5) †‡	<0.001	−0.223 (1.621)	−0.524 (1.436)	0.337
Appendicular muscle mass, kg	17.5 (3.6)	17.6 (3.7)	17.1 (3.7) †‡	<0.001	0.018 (0.740)	−0.302 (0.611)	0.047
SMI, kg/m^2^	6.7 (0.9)	6.8 (0.9)	6.6 (0.9) †‡	<0.001	0.005 (0.289)	−0.117 (0.240)	0.049
Body fat, kg	16.7 (6.4)	16.5 (6.4)	16.1 (6.5)	0.274	−0.177 (1.324)	−0.142 (1.762)	0.904
Percent body fat, %	27.6 (7.8)	27.4 (7.8)	27.7 (7.4)	0.692	−0.160 (1.799)	0.269 (1.856)	0.276

Mean (SD) of duration between point 1 and point 2 was 15.2 (5.1) months, and duration between point 2 and point 3 was 19.0 (4.3) months. * *p* value was evaluated among three points by repeated measures ANOVA. To compare the differences among three points, Bonferroni correction was used. † *p* < 0.05, vs. point 1 and ‡ *p* < 0.05, vs. point 2. Change between point 1 and point 2 was defined as (data at point 1–data at point 2)/duration (year). Change between point 2 and point 3 was defined as (data at point 2–data at point 3)/duration (year). ** *p* value was evaluated the differences between change from point 1 to point 2 and change from point 2 to point 3 by paired t test. SMI, skeletal muscle mass index.

**Table 3 ijerph-18-04188-t003:** Subgroup analysis of change in skeletal muscle mass index before and after the start of the COVID-19 pandemic.

		Point 1	Point 2	Point 3	* *p* Value	Change between Point 1 and Point 2	Change between Point 2 and Point 3	** *p* Value
Sex	Men, *n* = 35	7.1 (0.7)	7.2 (0.8)	7.0 (0.8) †‡	<0.001	0.031 (0.325)	−0.159 (0.257)	0.038
	Women, *n* = 21	6.0 (0.6)	6.0 (0.6)	5.9 (0.7)	0.301	−0.038 (0.217)	−0.048 (0.198)	0.885
Exercise habit	(−), *n* = 22	6.9 (0.7)	6.9 (0.7)	6.7 (0.7) †‡	0.018	0.001 (0.386)	−0.162 (0.289)	0.229
	(+), *n* = 34	6.6 (1.0)	6.6 (1.0)	6.5 (1.0) ‡	0.028	0.008 (0.212)	−0.088 (0.203)	0.088
Smoking	(−), *n* = 31	6.5 (0.9)	6.5 (0.9)	6.4 (0.9)	0.143	−0.000 (0.298)	−0.081 (0.256)	0.349
	(+), *n* = 25	7.0 (0.8)	7.1 (0.9)	6.8 (0.8) †‡	<0.001	0.012 (0.285)	−0.161 (0.219)	0.057
Alcohol	(−), *n* = 30	6.7 (0.9)	6.7 (1.0)	6.5 (1.0)	0.060	−0.025 (0.278)	−0.091 (0.256)	0.402
	(+), *n* = 26	6.8(0.8)	6.8 (0.9)	6.6 (0.8) †‡	0.003	0.040 (0.303)	−0.147 (0.224)	0.062
HbA1c	<7.0%, *n* = 28	6.7 (0.9)	6.7 (0.8)	6.7 (0.9)	0.405	−0.021 (0.297)	−0.064 (0.207)	0.570
	≥7.0%, *n* = 28	6.7 (0.8)	6.8 (1.0)	6.5 (0.9) †‡	<0.001	0.031 (0.285)	−0.170 (0.264)	0.042
Obesity	(−), *n* = 41	6.5 (0.7)	6.5 (0.8)	6.3 (0.8) †‡	<0.001	0.010 (0.297)	−0.133 (0.239)	0.067
	(+), *n* = 15	7.4 (0.9)	7.4 (0.8)	7.3 (0.7)	0.414	−0.009 (0.276)	−0.070 (0.249)	0.494
Duration	<20 years, *n* = 36	6.8 (1.0)	6.8 (1.0)	6.6 (1.0)	0.026	−0.037 (0.283)	−0.097 (0.248)	0.402
	≥20 years, *n* = 20	6.6 (0.6)	6.7 (0.8)	6.4 (0.6) †‡	0.007	0.082 (0.291)	−0.153 (0.229)	0.049

* *p* value was evaluated among three points by repeated measures ANOVA. To compare the differences among three points, Bonferroni correction was used. † *p* < 0.05, vs. point 1 and ‡ *p* < 0.05, vs. point 2. Change between point 1 and point 2 was defined as (data at point 1–data at point 2)/duration (year). Change between point 2 and point 3 was defined as (data at point 2–data at point 3)/duration (year). ** *p* value was evaluated the differences between change from point 1 to point 2 and change from point 2 to point 3 by paired *t* test.

**Table 4 ijerph-18-04188-t004:** Effect of COVID-19 pandemic on change in skeletal muscle mass index.

	*P*	η^2^
Time	0.029	0.108
Time * sex	0.511	0.009
Time * age	0.110	0.057
Time * BMI	0.535	0.008
Time * exercise	0.669	0.004
Time * smoking	0.839	0.001
Time * alcohol	0.517	0.009
Time * duration	0.138	0.048
Time * HbA1c	0.389	0.016

**Table 5 ijerph-18-04188-t005:** Subgroup analysis of change of skeletal muscle mass index before and after the start of the COVID-19 pandemic.

	Point 1	Point 2	Point 3	**p* Value	Change between Point 1 and Point 2	Change between Point 2 and Point 3	** *p* Value
Stress (−), *n* = 17	6.6 (0.8)	6.6 (0.9)	6.5 (0.9)	0.042	−0.013 (0.152)	−0.071 (0.155)	0.399
Stress (+), *n* = 5	6.7 (1.3)	6.9 (1.2)	6.8 (1.3)	0.402	0.202 (0.153)	−0.060 (0.296)	0.108
Sleep duration decrease (−), *n* = 18	6.5 (0.8)	6.6 (0.9)	6.5 (0.9)	0.302	0.032 (0.199)	−0.065 (0.217)	0.242
Sleep duration decrease (+), *n* = 4	7.0 (1.2)	7.1 (1.3) †	6.9 (1.3)	0.072	0.084 (0.032)	−0.082 (0.073)	0.005
Exercise decrease (−), *n* = 11	6.5 (0.5)	6.5 (0.5)	6.4 (0.7)	0.874	0.024 (0.205)	−0.027 (0.255)	0.643
Exercise decrease (+), *n* = 11	6.8 (1.2)	6.9 (1.2)	6.7(1.2) ‡	0.006	0.061 (0.154)	−0.107 (0.111)	0.036

* *p* value was evaluated among three points by repeated measures ANOVA. To compare the differences among three points, Bonferroni correction was used. † *p* < 0.05, vs. point 1 and ‡ *p* < 0.05, vs. point 2. Change between point 1 and point 2 was defined as (data at point 1–data at point 2)/duration (year). Change between point 2 and point 3 was defined as (data at point 2−data at point 3)/duration (year). ** *p* value was evaluated the differences between change from point 1 to point 2 and change from point 2 to point 3 by paired *t* test.

## Data Availability

The data that support the findings of this study are available on request from the corresponding author, Yoshitaka Hashimoto.

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
