# Peer review of "Effect of COVID-19 Pandemic on the Change in Skeletal Muscle Mass in Older Patients with Type 2 Diabetes: A Retrospective Cohort Study"

_ijerph, 2021, doi:10.3390/ijerph18084188_

Round 1

Reviewer 1 Report

This retrospective study by Hasegawa and others investigated the effect of the coronavirus disease (COVID-19) pandemic on the change in muscle mass in older patients with type 2 diabetes. The authors concluded that the COVID-19 pandemic caused muscle mass loss. Furthermore, the decrease in SMI after the pandemic was especially evident among the men, smokers, worse glycemic control, and those with a longer diabetes duration. The authors recommend prevent muscle mass loss; it will be necessary to recommend exercise to those at a higher risk sarcopenia.

This is very well written manuscript that draws attention the consequence of lock downs during a pandemic.  There are few areas where there needs to be clarification and improvement.

Abstract

The abstract conclusion needs to be expanded.  What is implication of these findings. Furthermore, the conclusion should be directed towards older adults with Type II diabetes.

Introduction

What would be the implication of losing muscle mass in a older adult population with Type II diabetes.  What are the primary and secondary aims of this study?

Methods

Were any functional measures taken in this study, such as, hand grip strength, sit to stand, or timed walking test?  These data would greatly improve this study.  If they are not available it should be listed as a limitation of the study.

Why was a paired T-test used for the change data when a T-test was used on the repeated measures data?

Discussion

It is this reviewer’s opinion, that the discussion could be improved by discussing the implications of losing skeletal muscle mass in this older adult population with Type II diabetes.

The conclusion should mention the population that was studied (diabetic older adults)

Author Response

Response to Reviewer-1

  1. Abstract

The abstract conclusion needs to be expanded. What is implication of these findings. Furthermore, the conclusion should be directed towards older adults with Type II diabetes.

Response

Thank you for your suggestion. According to your suggestion, we have revised the Abstract conclusions described as below.

The COVID-19 pandemic restrictions caused muscle mass loss in older patents with T2D. Actions, including recommendation of exercise and adequate diet intake, are needed to prevent loss of muscle mass.

  1. Introduction

What would be the implication of losing muscle mass in a older adult population with Type II diabetes. What are the primary and secondary aims of this study?

Response

Thank you for your comment. Muscle is a major organ for glucose metabolism; therefore, muscle is an important target for treatment of people with T2D. People with diabetes have accelerated muscle catabolism because of insulin signal attenuation and insulin resistance. Moreover, older patients with T2D often have sarcopenia. Therefore, for older people with T2D, prevention of muscle mass reduction is an important goal. The primary aim of this study was to investigate the effect of COVID-19 pandemic restrictions on the change in muscle mass in older patients with T2D and secondary aim was to clarify the risk factor for the change in muscle mass during COVID-19 pandemic restrictions. According to your comment, we have added these points in the Introduction section described as below.

Muscle is a major organ for glucose metabolism [9]; therefore, muscle is an important target for treatment of patients with T2D. Patients with diabetes have accelerated muscle catabolism because of insulin signal attenuation and insulin resistance [10]. Moreover, older patients with T2D often have sarcopenia [11]. Therefore, for older patients with T2D, prevention of muscle mass reduction is an important goal.

Thus, the primary aim of this retrospective cohort study was to investigate the effect of COVID-19 pandemic restrictions on the change in muscle mass in older patients with T2D who were not infected with COVID-19 and secondary aim was to clarify the risk factor for the change in muscle mass during COVID-19 pandemic restrictions.”

  1. DeFronzo RA, Tripathy D. Skeletal muscle insulin resistance is the primary defect in type 2 diabetes. Diabetes Care 2009; 32 Suppl 2: S157-S163.
  2. Umegaki H. Sarcopenia and frailty in older patients with diabetes mellitus. Geriatr Gerontol Int 2016; 16: 293-299.
  3. Okamura T, Miki A, Hashimoto Y, et al. Shortage of energy intake rather than protein intake is associated with sarcopenia in elderly patients with type 2 diabetes: A cross-sectional study of the KAMOGAWA-DM cohort. J Diabetes 2019; 11: 477-483.

  1. Methods

Were any functional measures taken in this study, such as, hand grip strength, sit to stand, or timed walking test? These data would greatly improve this study. If they are not available it should be listed as a limitation of the study.

Response

Thank you for your valuable comment. As you say, functional measures, such as hand grip strength, sit to stand, and timed walking test, are important and useful markers. Unfortunately, however, we did not have the data of these functional measures. According to your comment, we have mentioned this point as one of the limitations of this study descried as below.

“Fourth, functional measures, such as hand grip strength, sit to stand, and timed walking test are important markers. Unfortunately, however, we did not have the data of these functional measures.”

  1. Why was a paired T-test used for the change data when a T-test was used on the repeated measures data?

Response

Thank you for your comment. In this study, we compared the data of the same participants. Thus, differences among three points measurements were evaluated by repeated measure ANOVA and the changes of data were evaluated by paired T-test.

  1. Discussion

It is this reviewer’s opinion, that the discussion could be improved by discussing the implications of losing skeletal muscle mass in this older adult population with Type II diabetes.

Response

Thank you for your valuable suggestion. As you say, to discuss the implications of losing skeletal muscle mass in older adult with T2D could improve the discussion. According to your suggestion, we have added the sentences in the Discussion section described as below.

Muscle is a major organ for glucose metabolism [9]; therefore, muscle is an important target for treatment of patients with T2D. In fact, it has been reported that muscle mass loss is associated with low quality of life [29], cardiovascular disease [30,31] and mortality [32].

  1. DeFronzo RA, Tripathy D. Skeletal muscle insulin resistance is the primary defect in type 2 diabetes. Diabetes Care 2009; 32 Suppl 2: S157-S163.
  2. Tsekoura M, Kastrinis A, Katsoulaki M, Billis E, Gliatis J. Sarcopenia and Its Impact on Quality of Life. Adv Exp Med Biol 2017; 987: 213-218.
  3. Lai S, Muscaritoli M, Andreozzi P, et al. Sarcopenia and cardiovascular risk indices in patients with chronic kidney disease on conservative and replacement therapy. Nutrition 2019; 62: 108-114.
  4. Hashimoto Y, Kaji A, Sakai R, Hamaguchi M, Okada H, Ushigome E, et al. Sarcopenia is associated with blood pressure variability in older patients with type 2 diabetes: A cross-sectional study of the KAMOGAWA-DM cohort study. Geriatr Gerontol Int 2018; 18: 1345-1349.
  5. Miyake H, Kanazawa I, Tanaka KI, Sugimoto T. Low skeletal muscle mass is associated with the risk of all-cause mortality in patients with type 2 diabetes mellitus. Ther Adv Endocrinol Metab 2019; 10: 2042018819842971.

  1. The conclusion should mention the population that was studied (diabetic older adults)

Response

Thank you for your suggestion. According to your suggestion, we have mentioned the population that was studied described as below.

“This study revealed that the COVID-19 pandemic restrictions caused muscle mass loss in older patents with T2D. A decrease in SMI after the pandemic was especially remarkable among the men, worse glycemic control, and those with a longer diabetes duration. Actions, including recommendation of exercise and adequate diet intake, are needed to prevent loss of muscle mass.”

Reviewer 2 Report

Appreciate authors for attempting and reporting the impact of pandemic on muscle mass in patients with diabetes. The topic is interesting and still studies are warranted to emphasize the impact of lockdown or home-isolation on several lifestyle activities, physiological functions and biomarkers, especially in older adults with comorbidities. In this study, authors reported loss of skeletal muscle mass in diabetic patients due to long-time home stay or physical inactivity caused by the COVID-19 pandemic restrictions. Although the study findings are interesting, there are several concerns that should be addressed prior to finalize the article for publication.

Comments:

  1. From the Abstract it is not clear whether the diabetic patients were infected with COVID-19 or not. Please make it clear.
  2. Abstract conclusions: Authors mentioned that COVID-19 pandemic caused muscle mass loss. This study did not measured the muscle mass in CVOID-19 infected patients. Therefore, conclusions and main finding should revise to refer that pandemic-related physical inactivity may contribute to loss of muscle mass.
  3. Line 51-53: It is not clear whether diabetic patients were infected with the disease or not. The sentence should give the meaning ‘changes of muscle mass in physically inactive diabetic patients during the pandemic lockdown or home-stay’.
  4. Line 40-42: It is well-known that COVID-10 impacted lifestyle of all ages in all aspects. Here in authors are required to specify what lifestyle has dramatically changed (Ref. 5-7) and what was reported in their previous studies (Ref. 8).
  5. Line 59: seems incomplete sentence. Line 61: Please specify what data were obtained from the patients.
  6. Under the subtitle 2.1., please include the mean age and body weights of the patients recruited for this study.
  7. Table 1: Several details are missing, including units, detailed footnotes, abbreviations and some other descriptions.
  8. It is obvious that physical inactivity could lead to decrease the skeletal muscle mass. In this study, authors reported decreased muscle mass during the pandemic lockdown period. Other than inaccessible to exercise facilities or forced home-stay, what could be the interesting point to emphasize from this study?
  9. Table 1 and Table 5: Suggested to merge Table 1 and 5 to one table by adding one extra row and present the values.
  10. Line 191-192: Again ‘pandemic caused muscle mass loss’. Since there is no direct evidence that COVID-19 caused loss of skeletal muscle mass, authors should revise this sentence. Authors may use ‘pandemic physical inactivity’ or other suitable wording.
  11. Authors could discuss the association between other risk factors (for instance gender, smoking, alcohol, impaired glycemic control) and SMI in patients with diabetes.
  12. Do authors measured the physical activity levels of patients before pandemic? If so, authors could compare the data with post-pandemic data, which is reasonable to explain the decreased physical activity levels during home-stay.
  13. It would be very interesting to continue the study for few more weeks and do the exercise intervention.

Author Response

Response to Reviewer-2

  1. From the Abstract it is not clear whether the diabetic patients were infected with COVID-19 or not. Please make it clear.

Response

Thank you for your comment. The participants of this study were not infected with COVID-19. According to your comment, we have revised this point described as below.

Aim of this study was to investigate the effect of the coronavirus disease (COVID-19) pandemic restrictions on the change in muscle mass in older patients with type 2 diabetes (T2D), who were not infected with COVID-19.”

  1. Abstract conclusions: Authors mentioned that COVID-19 pandemic caused muscle mass loss. This study did not measured the muscle mass in CVOID-19 infected patients. Therefore, conclusions and main finding should revise to refer that pandemic-related physical inactivity may contribute to loss of muscle mass.

Response

Thank you for your comment. As you say, this study did not measure the muscle mass in CVOID-19 infected patients. COVID-19 pandemic restrictions during the pandemic lockdown or home-stay caused increases stress, changes in dietary patterns and decreased physical activity. According to your comment, we have revised the Abstract and Introduction sections described as below.

“The COVID-19 pandemic restrictions caused muscle mass loss in older patents with T2D.”

“Taking these findings together, COVID-19 pandemic restrictions during the pandemic lockdown or home-stay caused increased stress, changes in dietary patterns and decreased physical activity.”

  1. Line 51-53: It is not clear whether diabetic patients were infected with the disease or not. The sentence should give the meaning ‘changes of muscle mass in physically inactive diabetic patients during the pandemic lockdown or home-stay’.

Response

Thank you for your valuable suggestions. As you say, it is not clear whether patients with T2D were infected with COVID-19 or not. According to you and reviewer 1’s comment, we have changed the sentence described as below.

“Thus, the primary aim of this retrospective cohort study was to investigate the effect of COVID-19 pandemic restrictions on the change in muscle mass in older patients with T2D who were not infected with COVID-19 and secondary aim was to clarify the risk factor for the change in muscle mass during COVID-19 pandemic restrictions.”

  1. Line 40-42: It is well-known that COVID-19 impacted lifestyle of all ages in all aspects. Here in authors are required to specify what lifestyle has dramatically changed (Ref. 5-7) and what was reported in their previous studies (Ref. 8).

Response

Thank you for your comment. The COVID-19 pandemic impacted lifestyle, such as taking unhealthy dietary patterns, increasing alcohol consumption, increasing smoking frequency and reducing physical activity. In addition, our study revealed that COVID-19 pandemic increasing stress, increasing total diet intake, increasing snack consumption, decreasing sleep duration and reducing physical activity. According to your comment, we have added these points described as below.

“Thus, along with its spread, COVID-19 pandemic has led to a dramatic change in lifestyles, such as taking unhealthy dietary patterns, increasing alcohol consumption, increasing smoking frequency and reducing physical activity [5-7]. During the COVID-19 pandemic, our research group conducted a study of patients with type 2 diabetes (T2D) and reported that COVI-19 pandemic changes in the lifestyles, including increasing stress, increasing total diet intake, increasing snack consumption, decreasing sleep duration and reducing physical activity [8].”

  1. Line 59: seems incomplete sentence. Line 61: Please specify what data were obtained from the patients.

Response

Thank you for your comment. According to your comment, we have revised them described as below.

“All participants were selected from this cohort.”

“In this retrospective cohort study, data of BIA, lifestyle, medications, and laboratory data were obtained from outpatients with T2D who underwent bioelectrical impedance analysis (BIA) at least twice before April 2020 and at least once thereafter.”

  1. Under the subtitle 2.1., please include the mean age and body weights of the patients recruited for this study.

Response

Thank you for your comment. We have added mean age and body weight of the participants recruited for this study in Results section described as below.

“Mean (SD) age, body weight and duration of diabetes were 75.2 (7.1) years, 59.3 (10.3) kg and 19.7 (8.2) years, respectively.”

  1. Table 1: Several details are missing, including units, detailed footnotes, abbreviations and some other descriptions.

Response

Thank you for your comment. According to your comment, we have revised Table1.

Table 1. Baseline characteristics (point 2) of study participants.

N = 56

Point 1

Point 2

Point 3

Age, year

-

75.2 (7.1)

-

Men

-

35 (62.5)

-

Duration of diabetes, year

-

19.7 (8.2)

-

Smoker

-

25 (44.6)

-

Exercise

-

34 (60.7)

-

Alcohol consumer

-

26 (46.4)

-

Height, cm

-

160.5 (8.2)

-

HbA1c, %

7.2 (0.8)

7.1 (0.7)

7.1 (0.7)

HbA1c, mmol/mol

55.1 (9.1)

54.4 (7.9)

54.4 (7.9)

Insulin secretagogues

49 (86.0)

49 (86.0)

51 (89.5)

Insulin sensitizers

23 (40.4)

24 (42.1)

28 (49.1)

α-glucosidase inhibitor

11 (19.3)

11 (19.3)

13 (22.8)

Sodium glucose cotransporter 2 inhibitors

10 (17.5)

16 (28.1)

19 (33.3)

GLP-1 receptor agonist

3 (5.2)

4 (7.0)

5 (8.8)

Insulin

14 (24.6)

14 (24.6)

15 (26.3)

Renin-angiotensin system inhibitor

-

34 (59.6)

-

Calcium channel blocker

-

19 (33.3)

-

The other antihypertension drug

-

15 (26.3)

-

Medication for dyslipidemia

-

33 (57.8)

-

Stress increased (N = 22)

-

5 (22.7)

-

Sleep duration decreased (N = 22)

-

4 (18.1)

-

Exercise decrease (N = 22)

-

11 (50.0)

-

Insulin secretagogues includes sulfonylurea, dpp4 inhibitor and glinide. Insulin sensitizers included biguanides and thiazolidine. The other antihypertension drug included α-blocker, β-blocker and diuretic. Medication for dyslipidemia included statin, fibrate, eicosapentaenoic acid and ezetimibe. Data were express as means (standard deviation) and frequencies (percentage). GLP-1, glucagon like peptide-1.

  1. It is obvious that physical inactivity could lead to decrease the skeletal muscle mass. In this study, authors reported decreased muscle mass during the pandemic lockdown period. Other than inaccessible to exercise facilities or forced home-stay, what could be the interesting point to emphasize from this study?

Response

Thank you for your comment. As you say, physical inactivity could lead to decrease the skeletal muscle mass. In addition, change of diet intake also could lead to decrease the skeletal muscle mass. Therefore, COVID-19 pandemic restrictions caused the decreasing the skeletal muscle mass, theoretically. However, no previous studies reported that whether COVID-19 pandemic restrictions actually caused the decreasing the skeletal muscle mass or not. Thus, we believe this study has important meaning.

  1. Table 1 and Table 5: Suggested to merge Table 1 and 5 to one table by adding one extra row and present the values.

Response

Thank you for your suggestion. We have merged Table 1 and 5 to one table.

  1. Line 191-192: Again ‘pandemic caused muscle mass loss’. Since there is no direct evidence that COVID-19 caused loss of skeletal muscle mass, authors should revise this sentence. Authors may use ‘pandemic physical inactivity’ or other suitable wording.

Response

Thank you for your suggestion. According to your suggestion, we have revised the sentence described as below.

“In this study we observed a significant decrease in SMI in older patitnets with T2D who were not infected with COVID-19, due to the COVID-19 pandemic restrictions compared with the SMI before the pandemic, suggesting that the pandemic restrictions caused muscle mass loss.”

  1. Authors could discuss the association between other risk factors (for instance gender, smoking, alcohol, impaired glycemic control) and SMI in patients with diabetes.

Response

Thank you for your comment. It has been reported that poor glycemic control results in more muscle mass loss. It has been reported that among older patients with T2D, the decrease in muscle mass compared to non-diabetic people is more remarkable in men. Moreover, the longer the duration of diabetes, the more diabetic complications develop, accelerating muscle weakness and loss of muscle mass. According to your comment, we have added these points in the Discussion sections described as below.

“It has been reported that poor glycemic control results in more muscle mass loss [33-35]. It has been reported that among older patients with T2D, the decrease in muscle mass compared to non-diabetic people is more remarkable in men [36]. Moreover, the longer the duration of diabetes, the more diabetic complications develop, accelerating muscle weakness and loss of muscle mass [36, 37].”

  1. Lee CG, Boyko EJ, Barrett-Connor E, Miljkovic I, Hoffman AR, Everson-Rose SA, et al. Insulin sensitizers may attenuate lean mass loss in older men with diabetes. Diabetes Care 2011; 34: 2381-2386.
  2. Park SW, Goodpaster BH, Lee JS, Kuller LH, Boudreau R, de Rekeneire N, et al. Excessive loss of skeletal muscle mass in older adults with type 2 diabetes. Diabetes Care 2009; 32: 1993-1997.
  3. Sugimoto K, Ikegami H, Takata Y, Katsuya T, Fukuda M, Akasaka H, et al. Glycemic Control and Insulin Improve Muscle Mass and Gait Speed in Type 2 Diabetes: The MUSCLES-DM Study. J Am Med Dir Assoc 2020 in press.
  4. Park SW, Goodpaster BH, Strotmeyer ES, de Rekeneire N, Harris TB, Schwartz AV, Tylavsky FA, Newman AB. Decreased muscle strength and quality in older adults with type 2 diabetes: the health, aging, and body composition study. Diabetes. 2006 Jun;55(6):1813-8
  5. Almurdhi MM, Reeves ND, Bowling FL, Boulton AJ, Jeziorska M, Malik RA. Reduced Lower-Limb Muscle Strength and Volume in Patients With Type 2 Diabetes in Relation to Neuropathy, Intramuscular Fat, and Vitamin D Levels. Diabetes Care. 2016 Mar;39(3):441-7.

  1. Do authors measured the physical activity levels of patients before pandemic? If so, authors could compare the data with post-pandemic data, which is reasonable to explain the decreased physical activity levels during home-stay.

Response

Thank you for your comment. As you say, by measuring the amount of activity, we can explain the decrease in physical activity level during the home-stay. Unfortunately, however, we did not measure the physical activity levels of patients before pandemic.

Thus, we have mentioned this point as one of the limitations of this study described as below.

“Second, we were unable to quantitatively assess changes in engagement in physical activity, which is reasonable to explain the decreased physical activity levels during home-stay.”

  1. It would be very interesting to continue the study for few more weeks and do the exercise intervention.

Response

Thank you for your suggestion. As you say, to perform exercise intervention is interesting. We have mentioned this point in the Conclusions section described as below.

“Further studies would be interesting to perform the exercise intervention.”

Reviewer 3 Report

Thank you for giving me the opportunity to review the manuscript entitled “Effect of COVID-19 pandemic on the change in skeletal muscle mass in older patients with type 2 diabetes: a retrospective cohort study”. Due to the ongoing pandemic, the topic is timely and meaningful for health care.

Some comments for the authors include:

  • Main point: The authors conclude, that this study revealed that the COVID-19 pandemic caused muscle mass loss. Is there any information on the natural course? The authors compare data on muscle mass to a period before the pandemic, but the question is, follows the loss of muscle mass a linear function? Have the authors any data on the natural course of muscle loss in patients comparable to their cohort? Please comment!
  • The authors report that the decrease in SMI after the pandemic was especially remarkable among the men, smokers, worse glycemic control, and those with a longer diabetes duration. Might this be a serious confounder? Those with more serious disease suffered more loss of SMI? Please comment!
  • Results of BIA change with water balance (i.e. renal and cardiac failure) – no information is presented on this conditions. Please comment!
  • The number of included patients is very low compared to the entire study population. This might act as a confounder. Please comment.

Author Response

Response to Reviewer-3

  1. Main point: The authors conclude, that this study revealed that the COVID-19 pandemic caused muscle mass loss. Is there any information on the natural course? The authors compare data on muscle mass to a period before the pandemic, but the question is, follows the loss of muscle mass a linear function? Have the authors any data on the natural course of muscle loss in patients comparable to their cohort? Please comment!

Response

Thank you for your comment. As you say, whether the loss of muscle mass was a linear function after the pandemic and to compare our data with the data of natural course of muscle loss in patients with diabetes are important. Previous studies showed that muscle mass of patients with T2D is shown to be unchanged or slightly decreased in the natural course of the disease. In this study, muscle mass was unchanged before pandemic. On the other hand, their muscle mass was decreased after the pandemic. This might be due to the COVID-19 pandemic restrictions. According to your comment, we have added these points in the Discussion section described as below.

“Previous studies revealed that muscle mass of patients with T2D is shown to be unchanged or slightly decreased in the natural course [33-35]. In this study, muscle mass was unchanged before pandemic. On the other hand, their muscle mass was decreased after the pandemic. This might be due to the COVID-19 pandemic restrictions.

  1. Lee CG, Boyko EJ, Barrett-Connor E, Miljkovic I, Hoffman AR, Everson-Rose SA, et al. Insulin sensitizers may attenuate lean mass loss in older men with diabetes. Diabetes Care 2011; 34: 2381-2386.
  2. Park SW, Goodpaster BH, Lee JS, Kuller LH, Boudreau R, de Rekeneire N, et al. Excessive loss of skeletal muscle mass in older adults with type 2 diabetes. Diabetes Care 2009; 32: 1993-1997.
  3. Sugimoto K, Ikegami H, Takata Y, Katsuya T, Fukuda M, Akasaka H, et al. Glycemic Control and Insulin Improve Muscle Mass and Gait Speed in Type 2 Diabetes: The MUSCLES-DM Study. J Am Med Dir Assoc 2020 in press.

  1. The authors report that the decrease in SMI after the pandemic was especially remarkable among the men, smokers, worse glycemic control, and those with a longer diabetes duration. Might this be a serious confounder? Those with more serious disease suffered more loss of SMI? Please comment!

Response

Thank you for your valuable comment. We performed sub-analysis and showed that the patients with these cofounders were decreased their muscle mass due to the COVID-19 pandemic restrictions. As you say, the factors, such as men, worse glycemic control, and those with a longer diabetes duration, might be serious confounders. Thus, it has been reported that among older patients with T2D, the decrease in muscle mass compared to non-diabetic people is more remarkable in men. Moreover, the longer the duration of diabetes, the more diabetic complications develop, accelerating muscle weakness and loss of muscle mass. According to your comment, we have mentioned the effect of these cofounders in the Discussion section described as below.

It has been reported that poor glycemic control results in more muscle mass loss [33-35]. It has been reported that among older patients with T2D, the decrease in muscle mass compared to non-diabetic people is more remarkable in men [36]. Moreover, the longer the duration of diabetes, the more diabetic complications develop, accelerating muscle weakness and loss of muscle mass [36, 37].

  1. Park SW, Goodpaster BH, Strotmeyer ES, de Rekeneire N, Harris TB, Schwartz AV, Tylavsky FA, Newman AB. Decreased muscle strength and quality in older adults with type 2 diabetes: the health, aging, and body composition study. Diabetes. 2006 Jun;55(6):1813-8
  2. Almurdhi MM, Reeves ND, Bowling FL, Boulton AJ, Jeziorska M, Malik RA. Reduced Lower-Limb Muscle Strength and Volume in Patients With Type 2 Diabetes in Relation to Neuropathy, Intramuscular Fat, and Vitamin D Levels. Diabetes Care. 2016 Mar;39(3):441-7.

  1. Results of BIA change with water balance (i.e. renal and cardiac failure) – no information is presented on this conditions. Please comment!

Response

Thank you for your valuable comment. As you say, renal and cardiac failures are associated with the water balance. In this study, one participant had renal failure, defined by eGFR <30 mL/min/1.73 m2 and none of the participants had cardiac failure, defined by BNP ≥100 pg/ml. Thus, we have reanalyzed the data without the renal failure participant. The results were almost the same as those in original manuscript. According to your comment, we have revised the Methods and Results sections described as below.

“The exclusion criteria were as follows: 1) incomplete data, 2) data of point 3, the last BIA, conducted until June 2020, 3) period between point 1 and point 2 and/or point 2 and point 3 data collection of less than 6 months, 4) period between point 2 and point 3 data collection of more than 2.5 years, 5) non-elderly (age under 60 years old) patients [20], 6) patients with renal failure [21] and 7) patients with cardiac failure, defined by brain natriuretic peptide ≥100 pg/ml [22].

“Data on medications, including medications for diabetes, hypertension, and hyperlipidemia, and blood samples, including, creatinine and brain natriuretic peptide, were collected in the morning for biochemical measurements, were obtained from the medical records at point 2. In addition, the data of hemoglobin A1c (HbA1c) were collected at all points. The estimated glomerular filtration rate (eGFR; mL/min/1.73 m2) was calculated using the Japanese Society of Nephrology equation (eGFR = 194 × serum creatinine−1.094× age−0.287 × (0.739 for women)) [28]. Renal failure was defined as an eGFR of <30 mL/min/1.73 m2 [21].”

“Of the 105, we excluded 48 patients who met the exclusion criteria (Fig. 2): incomplete data (1 patient), data collected at point 3 till June 2020 (27 patients), data collection period between point 1 and point 2 and/or point 2 and point 3 which was less than 6 months (4 patients), data collection period between point 2 and point 3 which was more than 2.5 years (7 patients), age under 60 years old (6 patients), renal failure (one patient) and none of them had cardiac failure. Thus, 56 patients were finally included in this study.”

  1. Haneda M, Utsunomiya K, Koya D, Babazono T, Moriya T, Makino H, et al. A new Classification of Diabetic Nephropathy 2014: a report from Joint Committee on Diabetic Nephropathy. J Diabetes Investig 2015; 6: 242-246.
  2. Dickstein K, Cohen-Solal A, Filippatos G, McMurray JJ, Ponikowski P, Poole-Wilson P, et al. ESC Guidelines for the diagnosis and treatment of acute and chronic heart failure 2008: the Task Force for the Diagnosis and Treatment of Acute and Chronic Heart Failure 2008 of the European Society of Cardiology. Developed in collaboration with the Heart Failure Association of the ESC (HFA) and endorsed by the European Society of Intensive Care Medicine (ESICM). Eur Heart J 2008; 29: 2388-2442.
  3. Matsuo S, Imai E, Horio M, Yasuda Y, Tomita K, Nitta K, et al. Revised equations for estimated GFR from serum creatinine in Japan. Am J Kidney Dis 2009; 53: 982-992.

  1. The number of included patients is very low compared to the entire study population. This might act as a confounder. Please comment.

Response

Thank you for your comment. As you say, the number of included patients is low compared to the entire study population. This might act as a confounder. According to your comment, we have added this point as limitation of this study.

“Fifth, the number of included patients is low compared to the entire study population. This might act as a confounder.”

Reviewer 4 Report

Dear author

This study aimed to investigate the decline of skeletal muscle mass, SMI of Japanese patients with type 2 diabetes before and after the state of emergency during COVID-19 pandemic, and would be very important data for the treatment of DM patients in the future. However, several points should be re-analyzed and discussed as follows:

1)Was there significant difference in the period between from point 1 to point 2 and point 2 to point 3? Besides, if it had the difference, could it affect the decline of SMI before and after the state of emergency?

2)How many patients less than normal SMI based on AWGS [1], that 7.0kg/m2 in men and 5.7kg/m2 in women, were there in point 1, 2 and 3? Because the author is discussing the risk for sarcopenia according to the previous studies in discussion, this study should be discussed based on the above results.

[1]Liang-Kung Chen, Jean Woo, Prasert Assantachai, Tung-Wai Auyeung, Ming-Yueh Chou, Katsuya Iijima, et al. (2020). Asian Working Group for Sarcopenia: 2019 Consensus Update on Sarcopenia Diagnosis and Treatment. JAMDA,21(3),300-307. https://doi.org/10.1016/j.jamda.2019.12.012.

3)As you write in discussion, in patients with type 2 diabetes, the decline of SMI derived from inactivity could affect the control of the treatment for the patients, and it is very important issues. Therefore, the change of HbA1c and the prescriptions for drugs such as the rate of insulin secretagogues, insulin and others in point 2 and 3 of this study should be shown in results and discussed about the condition of real managements for the patients.

Sincerely yours,

From reviewer.

Author Response

Response to Reviewer-4

1)Was there significant difference in the period between from point 1 to point 2 and point 2 to point 3? Besides, if it had the difference, could it affect the decline of SMI before and after the state of emergency?

Response

Thank you for your comment. In this study, mean (SD) of duration between point 1 to point 2 was 15.2 (5.1) months and that between point 2 to point3 was 19.0 (4.3) months (p<0.001, by paired t test). As you say, there is a possibility that the difference in the period between from point 1 to point 2 and point 2 to point 3 might affect the decline of SMI before and after the state of emergency, although we corrected the effect of the difference in the period by determining the rate of change in SMI per year. Thus, we have mentioned this point in the Discussion section described as below.

“Third, there is a possibility that the difference in the period between from point 1 to point 2 and point 2 to point 3 might affect the decline of SMI before and after the state of emergency, although we corrected the effect of the difference in the period by determining the rate of change in SMI per year.”

2)How many patients less than normal SMI based on AWGS [1], that 7.0kg/m2 in men and 5.7kg/m2 in women, were there in point 1, 2 and 3? Because the author is discussing the risk for sarcopenia according to the previous studies in discussion, this study should be discussed based on the above results.

[1]Liang-Kung Chen, Jean Woo, Prasert Assantachai, Tung-Wai Auyeung, Ming-Yueh Chou, Katsuya Iijima, et al. (2020). Asian Working Group for Sarcopenia: 2019 Consensus Update on Sarcopenia Diagnosis and Treatment. JAMDA,21(3),300-307. https://doi.org/10.1016/j.jamda.2019.12.012.

Response

Thank you for your suggestion. As you say, to check the proportion of low muscle mass is interesting. According to your suggestion, we have investigated the proportion of low muscle mass. The proportion of sarcopenia was not changed from point 1 (n = 18) to point 2 (n = 18), whereas that were increased from point 2 (n = 18) to point 3 (n = 22). According to your suggestion, we have added this point in the Methods and Results sections described as below.

“In addition, low muscle mass was defined as SMI <7.0kg/m2 in men and <5.7kg/m2 in women [26].”

“In addition, the proportion of sarcopenia was not changed from point 1 (n = 18) to point 2 (n = 18), whereas that were increased from point 2 (n = 18) to point 3 (n = 22).”

  1. Chen LK, Woo J, Assantachai P, Auyeung TW, Chou MY, Iijima K, et al. Asian Working Group for Sarcopenia: 2019 Consensus Update on Sarcopenia Diagnosis and Treatment. J Am Med Dir Assoc 2020; 21: 300-307.

3) As you write in discussion, in patients with type 2 diabetes, the decline of SMI derived from inactivity could affect the control of the treatment for the patients, and it is very important issues. Therefore, the change of HbA1c and the prescriptions for drugs such as the rate of insulin secretagogues, insulin and others in point 2 and 3 of this study should be shown in results and discussed about the condition of real managements for the patients.

Response

Thank you for your suggestion. According to your suggestion, we have added the change of HbA1c and the prescriptions for drugs such as the rate of insulin secretagogues, insulin and others. Levels of HbA1c was not changed due to COVID-19 pandemic. On the other hand, usage of medications was increased. We have added these points in Methods, Results and Discussion sections described as below.

“Data on medications, including medications for diabetes, hypertension, and hyperlipidemia, and blood samples, including, creatinine and brain natriuretic peptide, were collected in the morning for biochemical measurements, were obtained from the medical records at point 2. In addition, the data of hemoglobin A1c (HbA1c) were collected at all points.”

“HbA1c was not changed due to COVID-19 pandemic restriction. On the other hand, the proportions of usage of insulin sensitizers and SGLT2 inhibitors were increased with COVID-19 pandemic restriction.”

“In addition, HbA1c was not changed due to COVID-19 pandemic restriction. On the other hand, the proportions of usage of insulin sensitizers and SGLT2 inhibitors were increased with COVID-19 pandemic restriction. These results suggest that glycemic control could have been done by adjusting medications.”

Round 2

Reviewer 2 Report

Appreciate authors for their detailed responses to each comment and careful corrections in the manuscript. The revised manuscript is suitable for publication. However English polishing is necessary.

  1. Lines 73-76: Please rephrase these sentences into two sentences. First sentence is about study design and second sentence is about data obtained from the participants. Please make sure the abbreviations…BIA.
  2. Appreciate authors for merging the Table 1 and 5 into one. In this connection, please revise the Table 1 in the revised manuscript.

Author Response

Response to Reviewer-2

  1. Lines 73-76: Please rephrase these sentences into two sentences. First sentence is about study design and second sentence is about data obtained from the participants. Please make sure the abbreviations…BIA.

Response

Thank you for your comment. According to your comment, we have revised line 73-76 as below.

“This study was a retrospective cohort study. Medical data were collected as anonymized data after obtaining informed consent and data of bioelectrical impedance analysis (BIA), lifestyle, medications, and laboratory data were obtained from the participants who underwent BIA at least twice before April 2020 and at least once thereafter in this study.”

  1. Appreciate authors for merging the Table 1 and 5 into one. In this connection, please revise the Table 1 in the revised manuscript.

Response

Thank you for your suggestion. According to your suggestion, we have revised the Table 1 as below.

N = 56

Point 1

Point 2

Point 3

Age, year

-

75.2 (7.1)

-

Men

-

35 (62.5)

-

Duration of diabetes, year

-

19.7 (8.2)

-

Smoker

-

25 (44.6)

-

Exercise

-

34 (60.7)

-

Alcohol consumer

-

26 (46.4)

-

Height, cm

-

160.5 (8.2)

-

HbA1c, %

7.2 (0.8)

7.1 (0.7)

7.1 (0.7)

HbA1c, mmol/mol

55.1 (9.1)

54.4 (7.9)

54.4 (7.9)

Insulin secretagogues

49 (86.0)

49 (86.0)

51 (89.5)

Insulin sensitizers

23 (40.4)

24 (42.1)

28 (49.1)

α-glucosidase inhibitor

11 (19.3)

11 (19.3)

13 (22.8)

Sodium glucose cotransporter 2 inhibitors

10 (17.5)

16 (28.1)

19 (33.3)

GLP-1 receptor agonist

3 (5.2)

4 (7.0)

5 (8.8)

Insulin

14 (24.6)

14 (24.6)

15 (26.3)

Renin-angiotensin system inhibitor

-

34 (59.6)

-

Calcium channel blocker

-

19 (33.3)

-

The other antihypertension drug

-

15 (26.3)

-

Medication for dyslipidemia

-

33 (57.8)

-

N=22

Age, year

-

75.3 (6.3)

-

Men

-

15 (68.1)

-

Duration of diabetes, year

-

21.2 (9.8)

-

Smoking

-

14 (63.6)

-

Exercise

-

14 (63.6)

-

Alcohol

-

10 (45.4)

-

Height, cm

-

155.1 (24.7)

-

HbA1c, %

-

7.4 (0.7)

-

HbA1c, mmol/mol

-

57.4 (7.6)

-

Insulin secretagogues

-

22 (100.0)

-

Insulin sensitizers

-

14 (63.6)

-

α-Glucosidase inhibitor

-

9 (40.9)

-

Sodium glucose cotransporter 2 inhibitors

-

7 (31.8)

-

GLP-1 receptor agonist

-

1 (4.5)

-

Insulin

-

6 (27.2)

-

Renin-angiotensin system inhibitor

-

15 (68.1)

-

Calcium channel blocker

-

3 (13.6)

-

The other antihypertension drug

-

5 (22.7)

-

Medication for dyslipidemia

-

13 (59.0)

-

Stress increased

-

5 (22.7)

-

Sleep duration decreased

-

4 (18.1)

-

Exercise decrease

-

11 (50.0)

-

Reviewer 3 Report

The authors adressed all suggestions carefully, thus the manuscript gained considerable quality. Congrats.

Author Response

Response to Reviewer-3

The authors adressed all suggestions carefully, thus the manuscript gained considerable quality. Congrats.

Response

We give deep thanks for your kind and detailed review.

Reviewer 4 Report

Dear author

 Thank you very much for your revised version.

I have one minor point I want you to revise into the following in the results.

“In addition, the proportion of low muscle mass was not changed from point 1 (n = 18) to point 2 (n = 18), whereas that were increased from point 2 (n = 18) to point 3 (n = 22).”

Because this study has not measured physical functions, such as grip strength and gait speed, the risk of sarcopenia can be predicted, but sarcopenia cannot be diagnosed.

Sincerely yours,

From reviewer.

Author Response

Response to Reviewer-4

I have one minor point I want you to revise into the following in the results.

“In addition, the proportion of low muscle mass was not changed from point 1 (n = 18) to point 2 (n = 18), whereas that were increased from point 2 (n = 18) to point 3 (n = 22).”

Because this study has not measured physical functions, such as grip strength and gait speed, the risk of sarcopenia can be predicted, but sarcopenia cannot be diagnosed.

Response

Thank you for your suggestion. As you say, we did not measure physical functions, such as grip strength and gait speed, thus, the risk of sarcopenia can be predicted, but sarcopenia cannot be diagnosed. According to your suggestion, we have revised the Results section described as below.

“In addition, the proportion of low muscle mass was not changed from point 1 (n = 18) to point 2 (n = 18), whereas that were increased from point 2 (n = 18) to point 3 (n = 22).”
